# The Use of the Static Posturography to Assess Balance Performance in a Parkinson’s Disease Population

**DOI:** 10.3390/ijerph20020981

**Published:** 2023-01-05

**Authors:** Sergio Sebastia-Amat, Juan Tortosa-Martínez, Basilio Pueo

**Affiliations:** Health, Physical Activity and Sports Technology (HEALTH-TECH), Department of General and Specific Didactics, Faculty of Education, University of Alicante, 03690 Alicante, Spain

**Keywords:** stabilometry, postural instability, aging, neurodegenerative

## Abstract

The literature has shown contradictory results so far about the use of posturography, especially static posturography, to evaluate balance performance in Parkinson’s disease (PD) populations. This study aimed to investigate the use of static posturography as a valid method to evaluate balance in a PD population. Fifty-two participants diagnosed with PD (Hoehn & Yahr stage: 1–3) were included in this cross-sectional study. All participants completed the following assessments: Hoehn and Yahr scale, Movement Disorder Society-Unified Parkinson’s Disease Rating Scale, Tinetti Scale, Berg Balance Scale, Activities-specific Balance Confidence scale, Timed Up and Go test, and Functional Reach Test. Sway parameters were analyzed with a baropodometric platform, under eyes open (EO) and eyes closed (EC) conditions, in a bipodal stance. Small to large correlations were observed between clinical balance tests and static posturography parameters, although the majority of these parameters correlated moderately. Considering posturographic variables, the highest correlation values were detected for total excursion (TE), mean velocity (MV), mean (X-mean), and root-mean-square (X-RMS) displacements in the medio-lateral directions. It was observed that posturographic parameters worsened as the disease progresses, although differences were only significant between the stages 1 and 3 in the H&Y scale (*p* < 0.05). Regarding the test condition, the visual deprivation worsened significantly all the static posturography parameters (*p* < 0.05), except the antero-posterior mean displacement (Y-Mean). Comparing visual conditions, the EC presented slightly higher correlation values with the clinical balance tests. Static posturography could be used as an objective complementary tool to clinical balance tests in order to assess and control balance performance, mainly to detect postural instability problems.

## 1. Introduction

Parkinson’s disease (PD) is a chronic and progressive neurodegenerative disease characterized by a wide range of motor complications [1,2]. Among them, postural instability is one of the most common and complex symptoms due to the unresponsiveness to the existing treatments [3]. Postural instability and resulting falls occur when patients lose the coordination of sensorimotor strategies to stabilize the body’s center of mass (COM) within the limits of stability imposed by the base of support (BOS) during activities involving weight transfer [4,5]. PD patients experience difficulties in balance with the ageing process, which will also be affected by the severity and duration of the disease [6]. Additionally, other factors related to PD may influence balance performance, such as stiffness, akinesia, gait disturbances, freezing of gait, cognitive impairment and abnormal anticipatory postural adjustments [3].

Postural instability detection is a clear goal in PD, since it is used for diagnosing and categorizing PD patients, not only in severity stages but also in subtypes classification based on phenotypes. Furthermore, postural instability assessment is important to detect and prevent future falls because of their association with fractures, poorer quality of life, reduced mobility, and social isolation [7,8,9].

There are different types of clinical tests to assess balance in PD patients [10]. Despite being validated, cheap, and easy to manage, these instruments do not offer direct measures about the underlying pathological process of postural instability in PD since the scoring system is based on the clinical subjective perception [11]. Hence, there is a need to find a method that assesses balance objectively, increasing the assessment accuracy and reducing the cost and time-consuming process.

Posturography seems to be an adequate method to evaluate objectively balance in PD patients. However, the literature has currently shown contradictory results [12,13]. Different studies found weak to strong correlations between clinical balance tests and static posturopgraphy parameters [14,15,16], whereas some studies observed no correlation [17], or the correlation was only found for dynamic posturography [13]. These discrepancies could be due to the dissimilar procedures employed in the posturographic studies, such as different trial duration, number of trials, resting time, foot positions, stabilometric parameters employed in the study, etc. The aforementioned differences, together with the wide variety of clinical balance measures used in the studies, are major constraints to extracting conclusions about the real use of static posturography. In order to standardize protocols and facilitate the comparison between stabilometric studies, we decided to use the most common stabilometric protocol employed in the literature based on the systematic review conducted by Kamieniarz et al. [12]. The variables that need to be considered in the measurement procedure are feet position, the number of trials, and the repetition time. The most standard procedure of each one of these variables independently according to this review is the double leg stance, three repetitions, and 30 s of repetition duration. However, this combination was only used in three studies [18,19,20], and a later one conducted recently by Kamieniarz et al. [21], none of which had the aim to observe the relationship between clinical balance tests and static posturography in PD. As far as we have been able to verify, this procedure has not been used previously for this aim. Therefore, the main purpose of this study was to investigate the use of static posturography as a valid method to evaluate balance performance in a PD population.

## 2. Materials and Methods

### 2.1. Participants

This is a cross-sectional study. The initial sample consisted of fifty-five participants with confirmed idiopathic PD, who were recruited from several Parkinson’s disease associations located in Comunidad Valenciana, Spain. Recruitment and data collection took place between the years 2018–2019. The inclusion criteria were as follows: participants diagnosed by a neurologist with idiopathic PD according to the clinical diagnostic criteria of the United Kingdom Parkinson’s Disease Society Brain Bank, above 18 years old, with a Hoehn & Yahr (H&Y) stage between 1–3, self-reported stable antiparkinsonian drugs treatment for at least 8 weeks before joining the study, capable of ambulating independently, able to give informed consent, and also follow simple instructions. Exclusion criteria were as follows: patients with a history of traumatic brain injury or stroke, severe chronic obstructive pulmonary disease, neurological disorders other than PD, myocardial infarction in the past 12 months, severe orthopedic problems in the lower limbs, and clinical diagnosis of dementia or severe cognitive impairment (Mini-Mental Status Examination Score or MMSE ≤ 24) [22]. Two people were detected as outliers and one person reported another neurological disease apart from Parkinson’s in the following 8 weeks after the testing session, so they were excluded from the study. Therefore, the final sample included fifty-two participants (Figure 1).

All tests were conducted with the participants in the “on” phase of their medication cycle, approximately 45–90 min after their morning dose of dopaminergic medication. It should be mentioned that all participants were assisted twice a week with physical therapy (one hour per session) in their respective PD associations. Before starting the study, participants were fully informed of the aim, benefits, and possible risks of their participation. The participants signed informed consent, previously approved by the research ethics committee of the University of Alicante (approval number: UA-2018-07-11), and all procedures were conducted in accordance with the Declaration of Helsinki.

### 2.2. Clinical Evaluation

Eligible participants were invited to meet with an interviewer who conducted a structured interview to compile sociodemographic and clinical information (Table 1). These data were used to carry out an initial screening. Subsequently, the participants were evaluated by a neurologist specialist in movement disorders who administered the Spanish motor section of the Movement Disorder Society-Unified Parkinson’s Disease Rating Scale (MDS-UPDRS III), which includes the H&Y scale. These scales were applied to assess PD motor symptomology and the stage of the disease (disability level), respectively. The same week, two psychologists administered the MMSE to evaluate cognitive function. All this information was used for the final screening.

### 2.3. Measurement Instruments of Balance for PD

Measurement instruments to assess balance in PD were selected according to the clinical and research relevance reported in the literature [10,23,24]. Therefore, the following tests were included:Tinetti scale “TS” (total score): It is a rating scale of 16 items (range 0–28 points) that assesses gait and balance performance, as well as fall risks [25,26]. It is composed of two subscales: Tinetti Gait Section “GS” (7 items, range 0–12 points) and Tinetti Balance Section “BS” (9 items, range 0–16 points). For this study, we analyzed TS and BS due to the aim of the research. Higher scores indicate better performance.Berg balance scale (BBS): The BBS is a rating scale of 14 items (range 0–56 points) that assesses balance performance and fall risk [27,28]. Higher scores indicate better performance.Activities-specific balance confidence scale (ABC scale): The ABC scale is a 16-item subjective questionnaire (range 0–100%) that rates the balance confidence of patients in activities of daily living [29]. The final score is obtained by adding the scores of individual items and then dividing by the total number. Higher scores mean good self-confidence.Timed up and go test (TUG): it is a validated and widely used tool for assessing functional mobility, dynamic balance, and as a predictor of falls in people with PD [30,31]. Participants were asked to rise from a chair, walk 3 m, turn, walk back and sit down, as quickly as they could safely without running. Higher time indicates worst performance.Functional reach test (FRT): The FRT was designed to assess dynamic balance performance. Participants were asked to reach the maximum distance with their arms outstretched in the forward direction while maintaining a fixed base of support [32,33]. The score of the test is the difference between the initial (arms at 90° shoulder flexion) and final position (maximum distance). A tape measure was located on the wall to measure the difference in centimeters. Higher distance indicates better performance.

Taking into account that the H&Y and the MDS-UPDRS III are considered gold standard tools for the evaluation of disease severity and PD symptomology, the authors decided to include them as balance instruments because of their relevance in the balance assessment.

The H&Y scale is commonly used to assess disease progression and classify patients according to disease severity. The scale ranged from stage 0 (absence of disease symptoms) to stage 5 (wheelchair mobility). Stage 1 is related with unilateral affectation, Stage 2 mild bilateral affectation, stage 3 mild to moderate bilateral affectation and stage 4 severe disability. The main difference between stage 2 and 3 is the presence of postural instability. Higher scores mean increased disease severity.The MDS-UPDRS is a reviewed version of the UPDRS scale, considered the gold standard in the clinical judgment of PD symptoms. In this case, we included only the motor section (part III) that ranged from 0 to 132. Higher scores mean increased motor affectation.

All participants were assessed by the same trained rater, except for the MDS-UPDRS III and the H&Y scale, which were conducted by the neurologist. The testing session included the static posturography, the TUG test, and the FRT. Taking the testing session as the reference (Figure 2), the clinical evaluation, as well as the balance scales, were performed previously at the same test conditions (hour, medication and location).

To ensure the stability of the measurements, three trials of motor clinically-based test (TUG and FRT) were conducted with 1 min rest between trials and 5 min between tests. The average score of the three trials was used for data analysis.

### 2.4. Static Posturography

Posturographic analysis was carried out using a baropodometric platform (FreeMed, Rome, Italy) with an active surface of 400 × 400 mm, 8 mm thickness, and a sampling frequency of 100 Hz. The participants were instructed to look straight ahead at an eye-level visual reference mark on a white wall placed 1.5 m in front of them. The feet were located apart at shoulder width and the arms rested alongside the body. In this position, the participants were asked to stand on the platform as still as possible until the researcher indicated the end of the trial. During the testing procedure, another researcher was located behind the participant to prevent falls.

Static balance was measured under eyes open (EO) and eyes closed (EC) conditions. The testing order was randomized. Three trials were conducted for each condition and the data collection period for each trial was 30 s with one-minute rest between trials [12]. The average score of the three trials was used for data analysis.

Stabilometric parameters measuring the deviation of the center of pressure (CoP) were: total excursion (TE), defined as the length of the total distance of the CoP during the trial duration (TE); area of the 95% confidence ellipse (CEA), defined as the smallest ellipse that will cover 95% of the points of the CoP diagram; mean velocity (MV), defined as total distance covered by the COP over the support base throughout the sampling time; and four parameters that measure average absolute displacements around the mean CoP: medio-lateral direction (X-Mean), antero-posterior direction (Y-Mean), root-mean-square amplitude in the medial-lateral direction (X-RMS), and root-mean-square amplitude in the antero-posterior direction (Y-RMS).

## 3. Statistical Analysis

Sample size calculation was performed based on a bivariate correlation model (two tails) using G*Power Software (version 3.1.9.7, University of Dusseldorf, Dusseldorf, Germany). A desired statistical power of 80%, a significance level of 0.05, and an expected correlation of 0.4 were set. The calculation revealed that at least 46 participants were required.

Statistical analyses were conducted using the Statistical Package for Social Sciences (v. 22, IBM, IBM Corp., Armonk, NY, USA). All variables included in this study were processed by Kolmogorov-Smirnov to check the criterion of normality and Levene’s test for the homogeneity of variance.

Descriptive variables are presented as mean and standard deviations. Posturographic parameters were compared across three groups (H&Y stage 1, H&Y stage 2, and H&Y stage 3) by using a one-way analysis of variance (ANOVA). When the ANOVA demonstrated significant differences, post hoc tests were performed (Tukey’s test or Games Howell test depending on the homogeneity of variances).

The association between static posturography and clinical balance tests was estimated using Pearson’s (*r*) or Spearman’s correlation coefficient (*ρ*), depending on the distribution of the data. The magnitude of correlation coefficients was interpreted as follows: trivial (<0.10), small (0.10–0.29), moderate (0.30–0.49), large (0.50–0.69), very large (0.70–0.90), and nearly perfect (>0.90) [34]. A paired t-test or Wilcoxon signed-rank test was used to compare differences between paired EO and EC conditions. The level of significance used was *p* < 0.05.

## 4. Results

### 4.1. Clinical Balance Instruments

The mean and standard deviation of balance instruments are depicted in Table 2. Considering the disease stage based on the H&Y scale, 28.85% of the participants showed clinical imbalance. In general, the clinical balance tests showed a low to moderate risk of falling. According to clinical balance tests, the majority of participants seems to be independent but presented difficulties to perform the different tasks required.

### 4.2. Posturographic Variables

Posturographic analysis showed that all variables worsened with visual deprivation. Significant differences were detected for all variables, except for the Y-Mean. In the case of X-Mean, a significant worsening was detected for the H&Y stages 2 and 3 when the test was conducted under EC conditions, but not for the H&Y stage 1. Analyzing the results by disease severity, significant differences were only detected between the H&Y stage 1 and 3 for all posturographic variables, except for the Y-Mean (Table 3).

### 4.3. Relationship between Clinical Balance Tests and Posturographic Variables

The results of the correlation analysis are shown in Table 4. Small to large correlations were found between the posturographic variables and clinical balance tests, although the majority of clinical balance tests correlated moderately. The correlation was significant for the majority of tests, except for the FRT, which also showed the lowest correlation values. Considering posturographic variables, the highest correlation was detected for the TE, MV, X-Mean, and X-RMS. Overall, posturographic analysis carried out under visual deprivation (EC) correlated better with clinical balance tests compared to baseline data (EO condition).

## 5. Discussion

The main purpose of this study was to investigate the use of static posturography as a valid method to assess balance performance. In general, our results showed that static posturography correlated moderately with clinical balance tests, showing higher correlations under EC conditions.

Inconsistent results have been found in the literature in this regard, with some studies [14,16,35] showing small to moderate correlations, according to our correlation threshold criteria. Conversely, other studies reported no significant correlation between clinical balance tests and static posturography parameters [13,17]. In the study of Souza et al. [13], the authors found significant correlations between clinical balance tests and dynamic posturography, but not to static posturography. They suggested that the majority of clinical balance tests required dynamic anticipatory postural adjustments, which are less involved in the static posturography assessment. However, the correlation values obtained in the present study between the clinical balance tests and the static posturography contradict these findings.

In our study, Tinetti (total score and balance section), BBS, and TUG showed the highest correlation with the posturographic variables, specifically with the TE, MV, X-Mean, and X-RMS. ABC scale also showed good correlation values but was slightly lower compared with the above. Similar to our results, Johnson et al. [14] reported that the strongest correlations were found with the TUG, BBS, and ABC scales (it should be noted that the Tinetti scale was not included in this study). The correlations values obtained for the Tinetti, TUG, and BBS seem sensible since these tests comprise static and dynamic components of the balance assessment [24] and demonstrated the strongest psychometric properties for the assessment of the balance [10], requirements to be considered an ideal balance tool. In the same way, the revision conducted by Kamieniarz et al. [12] highlighted these tests as the most commonly used clinical tests to assess balance performance in PD, and also in research to investigate the relationship between clinical balance tests and posturography because of their characteristics.

In regard to the disease stage, moderate correlations were found with the H&Y scale, probably due to the relevance of postural instability in the final score. In fact, balance disorders are diagnosed by clinicians with a score of 3 or higher in the H&Y scale [36]. Błaszczyk et al. [37] observed a significant correlation between the H&Y stage and different static posturography variables, although their correlation values were lower than ours. Otherwise, the MDS-UPDRS showed lower correlation values compared to the H&Y, possibly due to the inclusion of different items that are not directly related to the balance assessment. We observed worse values in all posturographic parameters as the disease progresses, although significant differences were observed only between the stage 1 and 3 for all posturographic variables, except for the Y-Mean. This makes sense since the participants classified as stage 1 are considered mildly affected whereas stage 3 are considered moderately affected but physically independent. Moreover, as we mentioned before, stage 3 is the first stage that exhibits clinical imbalance, so the presence of significantly worse stabilometric values is quite logical if we compare them with the less affected group. In this sense, Gimenez et al. [38] reported progressive balance deficits throughout the disease, being the differences significant when early PD stages were compared to stage 3 of the H&Y scale. Similarly, our results found significant differences between mild and moderate PD patients but not between early PD stages (H&Y stage 1 vs. stage 2). Recently, Kamieniarz et al. [21] observed significant differences in spatiotemporal COP measures when healthy controls and mild PD patients were compared to moderate PD patients. However, no differences were observed between healthy controls and mild PD patients, suggesting that spatiotemporal parameters could not detect balance impairments when these groups were compared. As an alternative, the authors proposed the use of advanced analytical methods of COP, such as sample entropy and power spectral density. Future studies are needed to confirm if the spatiotemporal parameters are capable of detecting balance differences between early PD stages of the disease, and also between early PD stages and healthy controls.

Considering test conditions, participants showed a deterioration in all posturographic parameters when the test was carried out under EC conditions, which is quite common even in healthy people [16]. The suppression of visual input involves a stoppage of information to the central nervous system that hinders participants to preserve postural control. There are different studies that conducted static posturography under EO and EC, although only a few studies discussed the visual dependency of PD during quiet stance [16,39,40,41,42,43,44], which showed no evidence for increased visual dependence in the PD population compared to a healthy older population. In our study, a significant worsening of all posturographic variables was observed except for the Y-Mean, regardless of disease stage. The fact that X-Mean and X-RMS worsened significantly when visual feedback was not allowed supports the hypothesis of Mitchell et al. [45] that PD patients tend to increase the medio-lateral sway as a strategy to reduce the antero-posterior sway, the most compromised for PD population. Consistent with our findings, previous studies showed that medio-lateral sway was more sensitive in detecting PD progression [46] and more predictive of future falls in PD [47].

## 6. Limitations

This study has potential limitations. The first is that participants were not divided according to PD subtypes, which could add information to understand the suitability of static posturography. Second, balance scales and motor clinically-based tests were applied on different days to avoid fatigue. Although tests were carried out under the same conditions (hour, medication intake, and location), the daily fluctuations of the PD symptomatology could influence the results. Future studies are required to confirm our results in a larger population.

## 7. Conclusions

Moderate correlations were obtained between clinical balance tests and posturographic variables, both under EO and EC conditions, although the EC condition showed slightly higher correlation values. The TE, MV, X-Mean, and X-RMS were the posturographic variables that reported the highest correlation with clinical balance tests. It was observed that posturographic parameters worsened as the disease progressed, although differences were only significant between stages 1 and 3 in the H&Y scale. Taken all together, static posturography could be used in clinical settings to help in balance assessments, providing objective information to health professionals. It could also be very useful as a complementary tool for monitoring disease progression. In the same way, because of the close relationship between balance impairments and falls, static posturography could offer relevant information to improve fall risk detection in PD patients.

## Figures and Tables

**Figure 1 ijerph-20-00981-f001:**
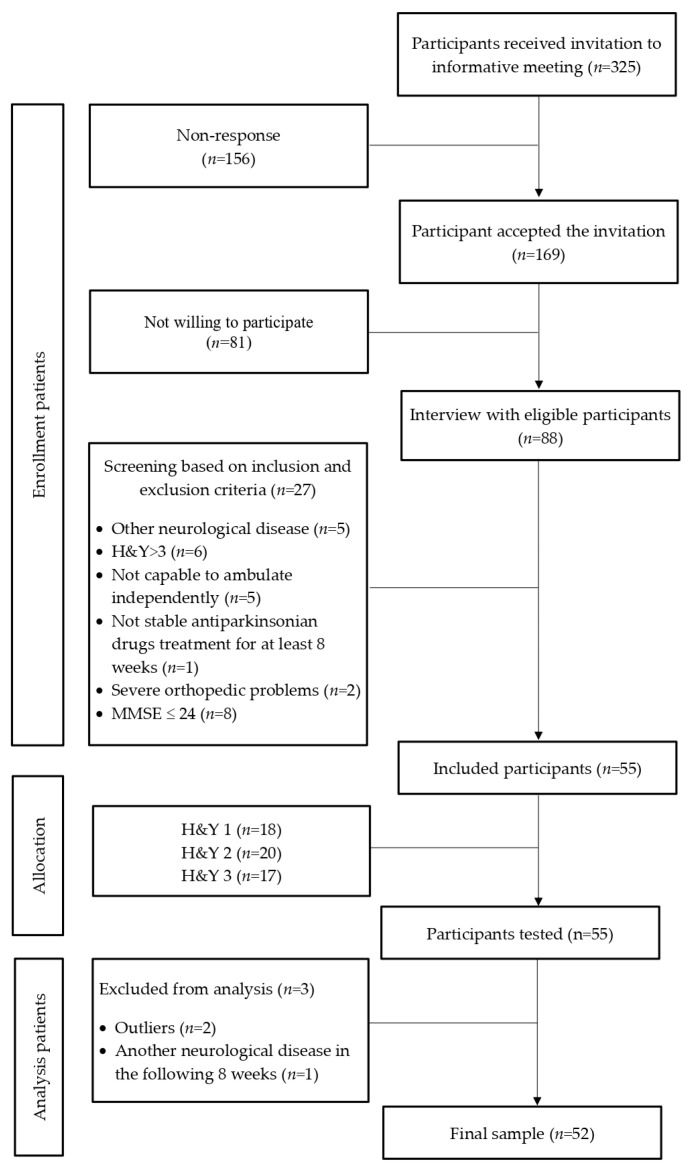
Flow diagram of the participants’ inclusion process.

**Figure 2 ijerph-20-00981-f002:**
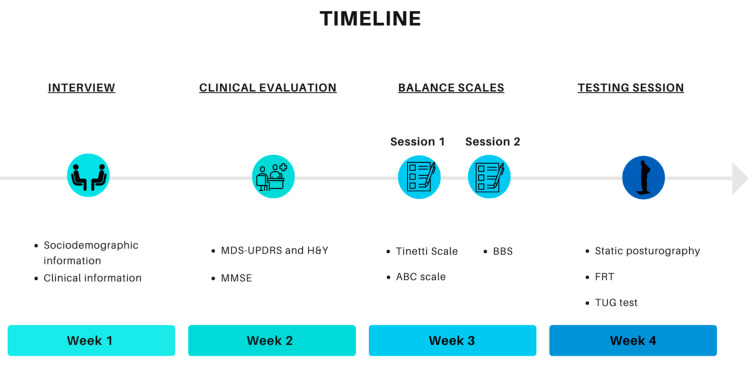
Data collection timeline.

**Table 1 ijerph-20-00981-t001:** Demographic characteristics.

Demographic Characteristics	Total Sample (*n* = 52)
Gender (men/women)	52 (36/16)
Age (years)	68.50 ± 8.37
Disease duration (years)	6.15 ± 4.14
BMI (kg/m^2^)	27.17 ± 5.82
Education (years)	10.54 ± 3.77
MMSE (score)	26.83 ± 2.31

BMI = body mass index; MMSE = mini-mental scale examination.

**Table 2 ijerph-20-00981-t002:** Clinical balance assessment instruments.

Balance Instruments	Total Sample (*n* = 52)
H&Y (1–5)	1.96 ± 0.79
*Stage 1 (n,%)*	17 (32.69%)
*Stage 2 (n,%)*	20 (38.46%)
*Stage 3 (n,%)*	15 (28.85%)
MDS-UPDRS-III (0–132)	31.73 ± 16.76
Tinetti (TS) (0–28)	23.02 ± 4.34
Tinetti (BS) (0–16)	13.69 ± 2.32
BBS (0–56)	48.19 ± 5.97
ABC scale (0–100%)	72.32 ± 16.19
TUG (s)	11.28 ± 4.84
FRT (cm)	20.57 ± 5.54

H&Y = Hoehn and Yahr scale; MDS-UPDRS (III) = modified unified Parkinson’s disease rating scale motor section; Tinetti (TS) = Tinetti total score; Tinetti (BS) = Tinetti balance score; BBS = Berg balance score; ABC = activities-specific balance confidence scale; TUG = timed up and go; FRT = functional reach test.

**Table 3 ijerph-20-00981-t003:** Descriptive data of posturographic variables. Differences between disease stages and visual conditions.

Posturographic Variables	Total Sample (*n* = 52)	H&Y Stage 1 (*n* = 17)	H&Y Stage 2 (*n* = 20)	H&Y Stage 3 (*n* = 15)	H&Y Stage 1 vs. H&Y Stage 3 *(p*-Value)
**TE (mm)**					
EO	458.56 ± 152.31	377.57 ± 105.25	467.23± 131.75	538.80 ± 179.18	0.006 **
EC	507.12 ± 185.12	406.98 ± 120.49	494.95 ± 143.24	636.82 ± 223.58	<0.001 **
*p*-value	<0.001 **	0.006 **	0.044 *	0.006 **	
**CEA (mm^2^)**					
EO	94.23 ± 158.57	51.32 ± 45.41	67.63 ± 48.07	178.32 ± 107.86	0.047 *
EC	144.62 ± 229.41	65.22 ± 59.33	101.16 ± 68.58	292.54 ± 225.66	0.011 *
*p*-value	0.022 *	0.144	0.047 *	0.016 *	
**MV (mm/s)**					
EO	15.61 ± 5.14	12.81 ± 3.60	15.82 ± 4.54	18.48 ± 5.93	0.004 **
EC	17.50 ± 7.27	13.69 ± 4.01	17.76 ± 5.87	21.45 ± 7.46	0.006 **
*p*-value	0.002 **	0.013 *	0.037 *	0.005 **	
**X-Mean (mm)**					
EO	7.14 ± 3.93	6.44 ± 3.83	7.02 ± 4.69	8.26 ± 4.76	0.035 *
EC	8.27 ± 4.22	6.91 ± 4.21	7.98 ± 4.64	9.62 ± 4.92	0.048 *
*p*-value	0.035 *	0.142	0.048 *	0.016 *	
**Y-Mean (mm)**					
EO	9.00 ± 6.23	8.45 ± 4.48	8.68 ± 6.22	10.03± 8.03	>0.05
EC	9.28 ± 6.90	9.45 ± 4.66	8.39± 6.96	10.27± 8.97	>0.05
*p*-value	0.412	0.052	0.653	0.714	
**X-RMS (mm)**					
EO	0.23 ± 0.08	0.18 ± 0.07	0.23 ± 0.07	0.28 ± 0.07	<0.001 **
EC	0.29 ± 0.13	0.21 ± 0.07	0.28 ± 0.14	0.38 ± 0.12	<0.001 **
*p*-value	<0.001 **	0.001 **	0.005 **	0.001 **	
**Y-RMS (mm)**					
EO	0.18 ± 0.05	0.16 ± 0.04	0.18 ± 0.04	0.21 ± 0.05	0.004 **
EC	0.22 ± 0.09	0.18 ± 0.05	0.22 ± 0.10	0.27 ± 0.10	0.015 *
*p*-value	<0.001 **	0.006 **	0.031 *	0.006 **	

TE = total excursion; CEA = area of 95% confidence ellipse; MV = mean velocity; X Mean = mean position in medial-lateral plane; Y Mean = mean position in antero-posterior plane; X-RMS: root mean square in antero-posterior direction; Y-RMS: root mean square in medial-lateral direction; EO = eyes open; EC = eyes closed; * = significant difference at *p* < 0.05; ** = statistically significant at *p*-value < 0.01.

**Table 4 ijerph-20-00981-t004:** Correlation coefficient (*r* or *ρ*) between clinical balance tests and posturographic variables.

	H&Y	MDS-UPDRS III	Tinetti (TS)	Tinetti (BS)	BBS	ABC	TUG	FRT
**TE (mm)**								
EO	0.45 **	0.34 *	−0.40 **	−0.36 **	−0.43 **	−0.43 **	0.45 **	−0.26
EC	0.52 **	0.43 **	−0.51 **	−0.48 **	−0.60 **	−0.46 **	0.52 **	−0.33 *
**CEA (mm^2^)**								
EO	0.38 **	0.18	−0.47 **	−0.48 **	−0.36 **	−0.31 *	0.36 **	−0.32 *
EC	0.42 **	0.30 *	−0.53 **	−0.60 **	−0.51 **	−0.29 *	0.40 **	−0.34 *
**MV (mm/s)**								
EO	0.48 **	0.36 **	−0.42 **	−0.38 **	−0.45 **	−0.45 **	0.44 *	−0.25
EC	0.52 **	0.42 **	−0.50 **	−0.49 **	−0.60 **	−0.48 **	0.52 **	−0.32
**X-Mean (mm)**								
EO	0.37 **	0.16	−0.46 **	−0.48 **	−0.33 *	−0.35 *	0.41 **	−0.31 *
EC	0.47 **	0.36 **	−0.55 **	−0.59 **	−0.54 **	−0.35 *	0.46 **	−0.26
**Y-Mean (mm)**								
EO	0.28	0.23	−0.44 **	−0.45 **	−0.33 *	−0.24	0.28	−0.22
EC	0.25	0.29 *	−0.38 **	−0.48 **	−0.33 *	−0.12	0.20	−0.28
**X-RMS (mm)**								
EO	0.56 **	0.43 **	−0.52 **	−0.45 **	−0.49 **	−0.47 **	0.48 **	−0.24
EC	0.62 **	0.53 **	−0.62 **	−0.60 **	−0.62 **	−0.50 **	0.51 **	−0.30
**Y-RMS (mm)**								
EO	0.38 **	0.42 **	−0.28	−0.28	−0.40 **	−0.20	0.31 *	0.09
EC	0.44 **	0.46 **	−0.34 *	−0.34 *	−0.46 **	−0.11	0.29 *	−0.15

H&Y = Hoehn & Yahr scale; MDS-UPDRS (III) = modified unified Parkinson’s disease rating scale motor section; Tinetti (TS) = Tinetti total score; Tinetti (BS) = Tinetti balance score; BBS = Berg balance score; ABC = activities-specific balance confidence scale; TUG = timed up and go; FRT = functional reach test; TE = total excursion; CEA = area of 95% confidence ellipse; MV = mean velocity; X Mean = mean position in the medial-lateral plane; Y Mean = mean position in the antero-posterior plane; X-RMS: root mean square in antero-posterior direction; Y-RMS: root mean square in medial-lateral direction; EO = eyes open; EC = eyes closed; * = statistically significant at *p*-value < 0.05; ** = statistically significant at *p*-value < 0.01.

## Data Availability

Data can be obtained through the corresponding author on reasonable request.

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
