# Peer review of "The Use of the Static Posturography to Assess Balance Performance in a Parkinson’s Disease Population"

_ijerph, 2023, doi:10.3390/ijerph20020981_

Round 1
Reviewer 1 Report
Certainly the authors have worked hard in preparing this manuscript, but
what important clinical purpose does this study have? What validity for doctors and physiotherapist?

Author Response
Response to Reviewer 1 Comments
Dear reviewer,
Thank you for your invaluable work that helps us to improve the quality of our research. Then, we are going to answer the different questions.
Comment 1 (Reviewer 1):
Certainly the authors have worked hard in preparing this manuscript, but what important clinical purpose does this study have? What validity for doctors and physiotherapist?
- Verses 65-66 “Therefore, the main purpose of this study was to investigate the use of static posturography as a valid method to evaluate balance performance in a PD population”.
What is the relevance of this study to clinical practice? What is the important contribution of a static test?
Response 1 (Reviewer 1):
Thank you for your comment. The clinical purpose of this study, as well as the validity for doctors and physiotherapists, are mentioned as follows:
- Balance disturbances, one of the main characteristics of Parkinson’s disease (PD), are usually diagnosed based on clinical instruments. These clinical tests do not offer direct measures about the underlying pathological process of postural stability since the scoring system is based on the clinician or physiotherapist subjective perception. This can lead to the occurrence of errors that involves transforming an appreciation into a numerical value.
The static posturography offer objective, direct and quantifiable measurements based on technology devices. The fact to provide objective measurements to health professionals could enable an accurate diagnosis, better monitoring of the disease progression and response to pharmacological or physical interventions, and also an improvement in the fall risk detection.
- Static posturography has been advanced in the literature as an alternative to evaluate balance problems. However, based on the systematic review conducted by Kamieniarz et al. (2018) contradictory results has been found about the suitability of this method to evaluate balance problems in PD patients. The authors of this review expose that these controversies could be due to the different methodologies employed in the studies (trial times, number of trials, etc.). Therefore, following the guidelines of these authors, the present study aimed to verify if posturography is an efficacy tool to analyze balance in people with Parkinson’s disease, comparing this method with standardized instruments for this condition.
- Finally, based on the literature, dynamic posturography seems to be more effective to evaluate balance and detect future falls in PD. However, in general terms, the cost to conduct this test is quite elevated compared to static posturography. The equipment is quite expensive (computerized dynamic posturography usually is usually up to ten times more expensive), the tests are more complex and need at least two professionals to conduct it (unless we have a harness, which is quite difficult to see in the clinical settings) and the interpretation of the results is more difficult. Given the above, the static posturography joins the best conditions to be employed both in the clinical setting and in the physiotherapist field.
If the reviewer 1 deems necessary to include an additional text in the conclusion section (or in another section) to explain the relevance of static posturography for clinician purpose, please let us know.
Comment 2 (Reviewer 1):
- 2.2. Clinical evaluation - superficially described.
Response 2 (Reviewer 1):
Thank you for your comment. More details have been added to the clinical section in order to improve understanding (lines 109-113).
Comment 3 (Reviewer 1):
- 2.4. Static posturography - Why were patients tested with three trials? Was it not tiring for them?
Were the scales used also used three times each?
Response 3 (Reviewer 1):
- Thank you for your comment. The use of three attempts is a quite usual procedure in the literature when motor tests are performed in neurodegenerative diseases, especially if a familiarization trial is not allowed. It is common to observe within subjects variation, so the use of a unique trial to statistical analysis could offer false results. Moreover, the review conducted by Kamieniarz et al. (2018) reported that three attempts were the most common used in the literature. In our case, we used the average value of the three trials.
- As we refer in the document, the only tests that were conducted three times were: static posturography (lines 186-187), the Timed Up and Go test (TUG) and the Functional Reach Test (FRT) (lines 168-169). In the case of the different balance scales (Tinetti, BBS and ABC) as well as the clinical scales (MDS-UPDRS and H&Y), these were applied only one time. In order to clarify testing order, the Figure 2 was created and included in the document.
- The static posturography, the Timed Up and Go test (TUG) and the Functional Reach Test (FRT) were conducted the same day (called testing session). It was not tiring for them since in the static posturography (3x30s) the participant only has to remain in a quite standing position with their feet located apart at shoulder width and the arms resting alongside the body. In the case of the TUG, the participants were asked to rise from a chair, walk 3 meters, turn, walk back and sit down (approximately 3x11-12 seconds each trial). Finally, in the FRT the participants were asked to reach the maximum distance with their arms outstretched in the forward direction while maintaining a fixed base of support (without moving the feet). This test was conducted in a few seconds.
It should be noted that all the participants were classified in the H&Y scale as 3 or below, which means that they were physically independents. Moreover, the participants were tested during the ON phase, which means that the medication is working optimally to reduce PD symptomatology.
Comment 4 (Reviewer 1):
- Description no limitations.
Response 4 (Reviewer 1):
Thank you for your pertinent comment. We included the limitations at the end of the discussion section (lines 331-337).
Comment 5 (Reviewer 1):
- This is a research article and not a review, so why did the authors refer to such old literature?
Response 5 (Reviewer 1):
The authors understand that the reviewer refers to balance scales references (the original article or validation of the test) or clinical balance tests to assess PD symptomatology. We consider that it is necessary to cite the original articles to provide information about these tests and to recognize the authorship.
- Folstein, M.F.; Folstein, S.E.; McHugh, P.R. “Mini-Mental State”: A Practical Method for Grading the Cognitive State of Patients for the Clinician. Journal of psychiatric research 1975, 12, 189–198, doi:10.3744/snak.2003.40.2.021.
- Kegelmeyer, D.A.; Kloos, A.D.; Thomas, K.M.; Kostyk, S.K. Reliability and Validity of the Tinetti Mobility Test for Individuals With Parkinson Disease. Physical therapy 2007, 87, 1369–1378, doi:https://doi.org/10.2522/ptj.20070007.
- Berg, K.O.; Wood-Dauphinee, S. L. Williams, J.I.; Maki, B. Measuring Balance in the Elderly: Validation of an Instrument. Canadian journal of public health 1992, 83, 7–11.
- Powell, L.E.; Myers, A.M. The Activities-Specific Balance Confidence (ABC) Scale. The Journals of Gerontology Series A: Biological Sciences and Medical Sciences 1995, 50, 28–34, doi:10.1093/gerona/50A.1.M28.
- Morris, S.; Morris, M.E.; Iansek, R. Reliability of Measurements Obtained With the Timed “Up & Go” Test in People With Parkinson Disease. Physical therapy 2001, 81, 810–818, doi:10.1093/ptj/81.2.810.
- Hoehn, M.M.; Yahr, M.D. Parkinsonism: Onset, Progression and Mortality. Neurology 1967, 17, 427–442.
In other cases, we included old literature, such as Mitchell et al. 1995, to explain hypothesis or physiological process postulated for some quite time, highlighting the work of these authors. This work is well recognized in the literature when we talk about mediolateral activity in PD patients.
- Mitchell, S.L.; Collin, J.J.; De Luca, C.J.; Burrows, A.; Lipsitz, L.A. Open-Loop and Closed-Loop Postural Control Mechanisms in Parkinson’s Disease: Increased Mediolateral Activity during Quiet Standing. Neuroscience Letters 1995, 197, 133–136, doi:10.1016/0304-3940(95)11924-l.
However, if we observe the literature referred to posturography, the majority of articles were published between 2011-2020. In fact, the systematic review cited in our article was published in 2018 (within the last four years).
- Kamieniarz, A.; Michalska, J.; Brachman, A.; Pawłowski, M.; Słomka, K.J.; Juras, G. A Posturographic Procedure Assessing Balance Disorders in Parkinson’s Disease: A Systematic Review. Clinical interventions in aging 2018, 13, 2301–2316, doi:10.2147/CIA.S180894.

Reviewer 2 Report
Title: The type of study carried out should be included, from what I have read, it is a descriptive cross-sectional observational study, joining this issue it is necessary to present strobe methodology for its realization.
Different balance and gait tests are presented, why are they carried out, as the study is on the posturograph.
As far as innovation is concerned, I am sorry to say that it is not new, as there are many studies on the subject.
Author Response
Response to Reviewer 2 Comments
Dear reviewer,
Thank you for your invaluable work that helps us to improve the quality of our research. Then, we are going to answer the different questions.
Comment 1 (Reviewer 2):
Title: The type of study carried out should be included, from what I have read, it is a descriptive cross-sectional observational study, joining this issue it is necessary to present strobe methodology for its realization.
Response 1 (Reviewer 2):
Thank you for this pertinent comment. We included the type of study, both in the abstract (line 11) and in the methods section (line 76).
Find attached the points that we included in the study according to the checklist of strobe methodology:
Strobe checklist
Title and abstract:
- Indicate the study’s design with a commonly used term in the title or the abstract (line 11).
Methods:
Participants
- Include the locations (line 78).
- Include periods of recruitment (lines 78-79).
- Give the eligibility criteria, and the sources and methods of selection of participants (Figure 1).
- Consider use of a flow diagram (Figure 1).
Study size
- Explain how the study size was arrived at (lines 200-203).
Comment 2 (Reviewer 2):
Different balance and gait tests are presented, why are they carried out, as the study is on the posturograph.
Response 2 (Reviewer 2):
In the case of posturography, the main application in PD population is to evaluate balance disorders, since it is one of the most common disturbances (Kamieniarz et al., 2018). Therefore, the present study aimed to verify if posturography is an efficacy tool to analyze balance in people with Parkinson’s disease, comparing this method with standardized instruments for this condition. The different balance instruments used in this study were selected according to the clinical and research relevance reported in the literature to evaluate balance performance (Bloem et al., 2016; Krzysztoń et al., 2018; Winser et al., 2019). The only included test that assess gait and functional mobility is the TUG test. However this is a test widely used to assess dynamic balance (Harris et al., 2015) and fall risk (Nocera et al., 2013).
Comment 3 (Reviewer 2):
As far as innovation is concerned, I am sorry to say that it is not new, as there are many studies on the subject.
Response 3 (Reviewer 2):
The authors agree with the reviewer that the use of static posturography to assess balance performance in PD patients is not new. However, based on the systematic review conducted by Kamieniarz et al. (2018) contradictory results has been found about the suitability of this method to detect balance problems in PD patients. The authors of this review expose that these controversies could be due to the different methodologies employed in the studies (trial times, number of trials, etc.). Curiously, even the 3x30s has been previously used to assess fall risk; it is not the case when the relationship between static posturography and standardized balance tests were observed. They usually use longer periods (51.2-60s), although these periods are less frequent in the literature than 30s. The same can be observed with the number of trials (a wide range: 1-8 trials) and the feet position (double leg, single leg and tandem stance) employed during the test. Therefore, following the guidelines of these authors to standardize protocols, the present study aimed to verify if posturography is an efficacy tool to analyze balance in people with Parkinson’s disease, comparing this method with standardized instruments.
References:
- Bloem, B. R., Marinus, J., Almeida, Q., Dibble, L., Nieuwboer, A., Post, B., ... & Movement Disorders Society Rating Scales Committee. (2016). M easurement instruments to assess posture, gait, and balance in P arkinson's disease: Critique and recommendations. Movement Disorders, 31(9), 1342-1355.
- Bloem, B. R., Visser, J. E., & Allum, J. H. (2003). Posturography. In Handbook of clinical neurophysiology(Vol. 1, pp. 295-336).
- Harris, D. M., Rantalainen, T., Muthalib, M., Johnson, L., & Teo, W. P. (2015). Exergaming as a viable therapeutic tool to improve static and dynamic balance among older adults and people with idiopathic Parkinson’s disease: a systematic review and meta-analysis. Frontiers in aging neuroscience, 7, 167.
- Kamieniarz, A., Michalska, J., Brachman, A., Pawłowski, M., Słomka, K. J., & Juras, G. (2018). A posturographic procedure assessing balance disorders in Parkinson’s disease: a systematic review. Clinical interventions in aging, 13, 2301.
- Krzysztoń, K., Stolarski, J., & Kochanowski, J. (2018). Evaluation of Balance Disorders in Parkinson's Disease Using Simple Diagnostic Tests—Not So Simple to Choose. Frontiers in neurology, 9, 932.
- Nocera, J. R., Stegemöller, E. L., Malaty, I. A., Okun, M. S., Marsiske, M., Hass, C. J., & National Parkinson Foundation Quality Improvement Initiative Investigators. (2013). Using the Timed Up & Go test in a clinical setting to predict falling in Parkinson's disease. Archives of physical medicine and rehabilitation, 94(7), 1300-1305.
- Winser, S. J., Kannan, P., Bello, U. M., & Whitney, S. L. (2019). Measures of balance and falls risk prediction in people with Parkinson’s disease: a systematic review of psychometric properties. Clinical rehabilitation, 33(12), 1949-1962.

Reviewer 3 Report
Dear Authors,
The article is quite complete in all sections and helps to enrich the literature findings. However, some details could be specified, as I suggest in the comments in the attached file, to make the work even more complete and reproducible.

Author Response
Response to Reviewer 3 Comments
Dear reviewer,
Thank you for your invaluable work that helps us to improve the quality of our research. Then, we are going to answer the different questions.
Introduction
Comment 1 (Reviewer 3):
Line 57 could you add (and briefly describe) in the introduction section literature references that use your chosen 30-second interval to perform posturography?
Response 1 (Reviewer 3):
Thank you for your comment. We pointed out this in the introduction section (line 60-63) based on the systematic review conducted by Kamieniarz et al. (2018). Moreover, as the reviewer 2 suggested, we included the studies that used this period of recording, as well as the feet position and the repetition duration (lines 63-69).
Methods
Comment 2 (Reviewer 3):
Can you specify in the methods section whether the participants did physical therapy or physical activity before or during the study?
Response 2 (Reviewer 3):
Thank you for your comment. We did not registered the physical activity before or during the study but all of the participants assisted twice a week to physical therapy in the PD associations (one hour per session).
We included this information in the methods section (lines 97-99).
Comment 3 (Reviewer 3):
Line 86 were the posturography and other “Measurement instruments of balance” performed on the same day? please add it in the methods section
Response 3 (Reviewer 3):
Thank you for your comment. We did not add this information and the authors agree with the reviewer that should be included in the methods section.
The static posturography, the Timed Up and Go test (TUG) and the Functional Reach Test (FRT) were performed on the same day (called Testing session). To avoid fatigue, in the case of the different balance scales (Tinetti, BBS and ABC), these were performed in different days (previous to Testing session). With the aim to clarify the study’s timeline, we added a figure in the methods section (Figure 2).
Comment 4 (Reviewer 3):
Line 164 you specified that Three trials were conducted for each condition. When you performed the statistical analysis to correlate the posturography with the different H&Y stages and the various balance tests, did you consider the average of the three trials for each condition or the best performance of the three trials in relation to the posturographic variables? you should specify that
Response 4 (Reviewer 3):
Thank you for this pertinent comment. We specified that we use the average score of the three trials in the TUG and FRT, but we did not specify this information for the static stabilometry. We also used the average of the three trials for the static stabilometry conducted in both test conditions. Therefore, we include this information in the document (line 188).
Comment 5 (Reviewer 3):
Line 217-218 Please specify at what level is the significance of the “#” symbol or add a column to table 3 after "H&Y stage 3" with the p-value between H&Y stage 1 and H&Y stage 3
Response 5 (Reviewer 3):
Thank you for your comment. We included an additional column to specify the significance (see Table 3).
References:
- Kamieniarz, A.; Michalska, J.; Brachman, A.; Pawłowski, M.; Słomka, K.J.; Juras, G. A Posturographic Procedure Assessing Balance Disorders in Parkinson’s Disease: A Systematic Review. Clinical interventions in aging 2018, 13, 2301–2316, doi:10.2147/CIA.S180894.

Round 2
Reviewer 1 Report
The authors have made corrections, but the manuscript needs further revisions.

Author Response
Response to Reviewer 1 Comments
Dear reviewer,
Thank you for your invaluable work that helps us to improve the quality of our research. Then, we are going to answer the different questions.
Comment 1 (Reviewer 1):
- The authors have made significant improvements.
Response 1 (Reviewer 1):
Thank you for your comment.
Comment 2 (Reviewer 1):
- Limitations - after discussion but as a separate item.
Response 2 (Reviewer 1):
Limitations were located in a separate item, after the discussion section (line 332).
Comment 3 (Reviewer 1):
- Conclusion - require significant improvements. What are the practical implications of this study? What is so important this study brings to clinical management?
Response 3 (Reviewer 1):
The practical implications to clinical management were included in the conclusion section (lines 346-351).

Reviewer 2 Report
Congratulations to the authors for their efforts, following the recommendations made by the reviewers.
Author Response
Response to Reviewer 2 Comments
Dear reviewer,
Thank you for your invaluable work that helps us to improve the quality of our research. We hope that these modifications are sufficient to improve this paper to merit publication in your Journal.
Kind Regards,
